# Spatio-Temporal Variation and Decomposition Analysis of Livelihood Resilience of Rural Residents in China

**DOI:** 10.3390/ijerph191710612

**Published:** 2022-08-25

**Authors:** Shulei Cheng, Yu Yu, Wei Fan, Chunxia Zhu

**Affiliations:** 1School of Public Administration, Southwestern University of Finance and Economics, Chengdu 611130, China; 2School of Economics, Southwestern University of Finance and Economics, Chengdu 611130, China

**Keywords:** livelihood resilience, rural residents, Dagum Gini coefficient, kernel density, spatial convergence

## Abstract

**Highlights:**

**What are the main findings?**
Livelihood resilience of rural residents (LRRR) in China was high in the east and low in the center and west.Inter-regional differences in LRRR were the main source of overall differences.σ and β convergence in LRRR were observed in most provinces.

**What is the implication of the main finding?**
A factual reference for policies related to reducing inter-provincial differences in the LRRR was provided.

**Abstract:**

The key to sustainable rural development and coordinated regional development is to properly measure the livelihood resilience of rural residents (LRRR), and investigate its regional differences, distribution characteristics, and evolutionary patterns. This study combined the entropy method, the Dagum Gini coefficient and decomposition, kernel density estimation, and convergence analysis to measure the LRRR in 30 provinces of China from 2006 to 2020, and to analyze its regional differences and sources, dynamic distribution, and characteristics of convergence. The LRRR in China overall declined 2006–2020, with an east-to-west spatial gradient toward lower livelihood resilience. Intra-regional differences in LRRR narrowed in the Eastern and Central Regions, while those in the Western Region widened. Inter-regional differences were the main source of differences in LRRR. The LRRRs in most provinces in China were gradually reaching the same level over time (i.e., σ convergence and β convergence). This research provides a factual reference for policies related to reducing inter-provincial differences in the LRRR in China.

## 1. Introduction

Livelihoods refer to the means and ways in which people use the resources available around them to maintain and enhance their basic living conditions [1,2]. To a certain extent, livelihood can represent the ability of residents to resist risks and external shocks. This means that setting indicators to quantitatively measure the ability of local residents to adapt and transform in both the changing social environment and the natural environment has important practical significance for promoting the sustainability of residents’ livelihoods. The livelihood resilience value can describe the resilience of residents in production and life after external shocks. Therefore, in the case of changes in the social environment and frequent natural disasters, livelihood resilience has gradually become an important reference standard for measuring the quality of life of local residents. At the same time, due to the unbalanced economic development between urban and rural areas in China, there will be obvious regional differences in the livelihood resilience of residents.

As urbanization continues to soar in China, urban and rural areas have taken different development paths in terms of economic structures and social networks, and therefore, urban and rural residents in China also have differing access to livelihoods [3]. While urban residents mostly rely on a combination of business, social security, and urban public services, which have high stability and sustainability, rural residents tend to sustain their livelihoods by using natural endowments, fixed assets, and social network resources [3,4]. Compared to rural residents, urban residents have higher levels of livelihood and better livelihood security [5]. This is mainly because, firstly, rural residents principally depend upon agricultural income, and the agricultural economy is extremely susceptible to environmental changes and human activities [6,7]. In general, rural residents have lower incomes, less social capital, and are less advantaged in the face of abrupt political or economic changes and devastating natural disasters than are urban residents, who have more opportunities for education and employment [4,8]. Secondly, economic and social policies are mostly formulated and implemented with a bias towards urban development. These policies mostly favor urban residents rather than rural residents, resulting in a huge gap between urban and rural residents in terms of income and social security [9].

Meanwhile, most local governments’ fiscal expenditures tend to be constructive expenditures that promote economic growth in the short term, while livelihood expenditures such as education and health care, which have a longer investment return cycle, are downplayed [10]. Biased economic and social policies and the fiscal expenditure structures of local governments have resulted in a wide gap between urban and rural basic public services, mainly in the form of poorer infrastructure and public service provision in rural areas that is inadequate to protect people’s livelihoods [5,11]. In addition, due to limited access to public resources, rural residents recover more slowly after natural disasters [7]. Thus, compared to urban residents, rural residents are disadvantaged in terms of livelihood capacity. However, although livelihood capacity can represent to some extent the ability of rural residents to withstand risks and external shocks, it cannot fully portray the ability of rural residents to recover in production economies and life events after being exposed to external shocks. Since the concept of livelihood resilience represents the ability to adapt and change in the changing society and natural environment, the livelihood resilience of rural groups is more worthy of attention.

One of China’s main challenges in implementing the “Rural Revitalization” strategy is how to construct better and more sustainable villages in the midst of urbanization [12]. The external impacts on rural dwellers are no longer limited to environmental change, but also to the ensuing socioeconomic changes. For sustainable rural development, it is crucial to quantify how well rural populations can adapt to changes in their social and natural environments. On this basis, this paper will offer policy suggestions for promoting the sustainability of rural residents’ livelihoods and for reducing regional differences by optimizing the index system and by clarifying the dynamic changes and regional differences affecting the livelihood resilience of rural populations.

## 2. Literature Review

Resilience provides a dynamic perspective for the study of rural residents’ livelihoods. The study of resilience first emerged in the fields of ecology and environmental studies. Resilience was defined as the ability of natural ecosystems to adapt to new environments and return to their original state after environmental disturbances or abrupt changes [13,14]. Resilience has four components—latitude, resistance, precariousness, and panarchy—in its effort to retain the same function, structure, identity, and feedbacks [15]. Since livelihoods represent systemic shifts and long-term changes, many scholars have transferred the concept of resilience to the field of livelihood studies, combining livelihoods and resilience [1,16]. On one hand, livelihood resilience can describe the extent to which livelihoods are affected and exposed to risks, shocks, and stresses, i.e., livelihood vulnerability [17,18]. On the other hand, livelihood resilience can express the adaptive capacity of livelihoods, i.e., the ability to diversify or shift livelihood strategies to absorb stresses and shocks through the mobilization and use of capital [19]. Therefore, most existing studies consider livelihood resilience as a portrayal of the ability of residents to maintain and enhance their resource endowment, well-being, and survival and return to their original livelihood state in the face of economic, social, environmental, and political changes [20,21]. Investigating livelihood resilience can enhance the dynamic understanding of the livelihood resilience of residents by linking livelihood vulnerability and adaptation [22,23]. Since rural residents may be more vulnerable to shocks and difficulties in returning to their original state during changes in the social environment or natural disasters [24], the investigation of the dynamic and differential presentation of livelihood resilience of residents should focus more on vulnerable rural groups than urban groups.

The literature shows that the dynamics of the livelihood resilience of rural residents (LRRR) are highly correlated with external pressures on rural residents, and with related policies. First, changes in the natural environment significantly affect LRRR. Some studies have found that frequent natural disasters such as earthquakes, floods, and droughts can reduce LRRR and widen the gap in livelihood resilience among rural residents [25,26]. However, by lowering agricultural productivity and eroding resistance, severe climate change can also limit LRRR [27,28]. Second, LRRR is highly correlated with regional macro policies. To address the multiple dilemmas of post-disaster reconstruction, environmental protection, improvement of human well-being, and alleviation of poverty, governments have targeted policies such as environmental migration and poverty alleviation relocation [2,29]. Among them, the environmental migration policies can improve the natural capital of rural residents, and the poverty alleviation relocation policies can improve their physical capital and human capital. Therefore, most of the policies can improve LRRR to varying degrees [30,31,32]. From these studies, it is easy to find that properly defining the meaning of LRRR, comprehensively measuring its level, and portraying its evolution pattern are not only key prerequisites for promoting the development of rural residents’ livelihoods, but also important and realistic references for assessing the effects of implementing existing relevant policies.

Numerous studies have analyzed the measurement of LRRR, but there is still no agreement over the optimal composition of its indicators. Most of the literature has characterized the idea of livelihood resilience in terms of two key issues: capacity and capital. Focusing on the capability perspective, Speranza et al. [22] developed a three-dimensional framework for analyzing livelihood resilience. The framework includes three types of indicators: buffer capacity, self-organization capacity, and learning capacity. Several studies have applied this analytical framework to examine changes in LRRR and concluded that buffer capacity is a key factor in maintaining the resilience of rural residents’ livelihoods [33,34]. Based on this, Quandt [21] drew on the sustainable livelihoods approach to further subdivide buffer capacity from a capital perspective. Capital was decomposed into financial capital, human capital, social capital, physical capital, and natural capital. This type of livelihood construction focuses more on the material value of capital [3,35], arguing that livelihood strategies and levels of well-being are heavily influenced by various levels of and access to livelihood capital [1,36]. Among them, for rural residents, social capital is more effective in coping with external changes in rural network relations [37]. Other works in the literature have studied the relationship between livelihood and risk, based on this framework, and found that both environmental risk and disease risk have a significant negative impact upon the achievement of sustainable livelihoods for rural residents [38]. The above studies have mainly focused upon the livelihood resources available to rural residents, but less attention has been paid to the impact of rural residents’ interactions with the external environment upon livelihood resilience. However, some scholars have constructed a new research framework that incorporates the adaptive capacity [23,39], or disaster resilience [7,40], of rural residents, arguing that the ability to choose livelihood strategies in response to changing external conditions is equally critical. On this basis, LRRR could be measured through disturbance, sensitivity, and adaptability at the farm level [41].

In summary, the existing literature provides insights into the study of the LRRR in China, but there is still room for expansion in terms of the indicator system and research content. First, previous studies have selected indicators based on the connotation and extension of LRRR, while mostly using survey data, and relatively few studies have used macro data to measure LRRR [40,42]. Because of this, the social security and public services provided by local governments to rural residents, which are important ways for rural residents to sustain their livelihoods in China, have also been de-emphasized when measuring LRRR. Given this, this study used macro data and extended prior research by integrating indicators of fiscal expenditure upon agriculture and the minimum living allowance for rural residents in order to measure the government’s support for rural residents’ livelihoods. An optimized system of indicators for measuring LRRR can provide an accurate and scientific assessment of the LRRR in each province of China. Existing research has mainly described and evaluated LRRR qualitatively through case studies, or measured LRRR using survey data. In the current study, the measurement indicators are comprehensive, specific, and more targeted [43], but this precision makes it difficult to compare LRRR across years and regions. In addition, to the best of our knowledge, no qualitative or quantitative descriptions of regional differences in LRRR have been made previously, nor have studies on the variation, spatial differences, and dynamic distribution of LRRR in China been conducted [31]. However, clarifying disparities in LRRR among different provinces and regions and measuring the catch-up momentum of provinces with lower levels of LRRR would help to understand the regional differences in China’s rural development and to formulate regionally targeted rural development policies. Given this, this paper uses a combination of the Dagum Gini coefficient and decomposition, kernel density estimation, and convergence analysis to dissect the evolutionary trends and regional differences in LRRR in China. The aim is to enhance the understanding of the current dynamics of the LRRR in China and to provide a reference for achieving sustainable development and coordinated regional development in rural areas.

## 3. Materials and Methods

### 3.1. Indicators of Livelihood Resilience of Rural Residents

Most of the existing literature on livelihood resilience has utilized research frameworks constructed from the perspectives of livelihood capacity and livelihood capital, in which the capacity-based livelihood resilience measures portray the social security and public services available to rural residents and show the local government’s support for rural residents’ livelihoods. Therefore, this study referred to Speranza’s study and adopted the three dimensions of buffer capacity, self-organization capacity, and learning capacity in order to assess the LRRR in China [22].

Buffer capacity consists mainly of a resource endowment of rural residents which has a buffer function against external changes [44]. It is usually expressed through physical capital, human capital, and financial capital. [33]. Referring to existing studies, this study selected livestock-rearing [45], possession of agricultural machinery, garden area, crop cultivation area [46], per capita income [45,47] and agricultural fixed asset investment [46,47] for measurement.

Self-organization capacity primarily reflects the ability of rural residents to integrate into the local economy and the social and institutional environment [46]. It is usually measured through government policy support, accessibility of transportation, cooperative relationships, and mutual trust among rural residents [48,49]. Since macro data were used to study the LRRR in this study, the interaction between rural residents and local society had to be elevated to the interaction of rural residents with local government. The main manifestation of this metric was the ability of rural residents to integrate into the local society and environment according to government expenditure on relevant domains, because it is the government and the community that provide rural residents with the opportunity to access or have the resources they need, and these resources serve to alleviate poverty and improve livelihoods [50]. In this study, the following variables were chosen for measurement: fiscal expenditure on agriculture, forestry, and water affairs [40]; minimum living allowance; medical care [40]; and postal delivery routes [26].

Learning capacity refers to the ability of rural residents to repair the negative effects of disturbances by acquiring knowledge, skills, and information and imitating, updating, and creating new things to improve their adaptability in a changing environment [29]. Learning capacity is usually measured by rural residents’ educational expenditure, education, work experience, and agricultural skills training [2,28]. This study used education expenditure [29] and the percentage of agricultural skills training as indicators of learning capacity.

The specific definitions and units of measurement of the indicators are shown in Table 1.

### 3.2. Entropy Method

Due to the different scales and multiple dimensions of the indicators measuring livelihood resilience, as outlined in Section 3.1, it is conventional to use the entropy method for calculations synthesizing a composite indicator. The entropy method is an approach for determining the weights of indicators and assigning them according to the degree of variation in each indicator value; it is widely used in economic and environmental research [51,52]. The entropy method largely avoids bias brought by human factors and has the advantage of objective assignment [53]. Therefore, this study adopted the entropy method to measure LRRR [45,47]. The specific calculation steps were as follows:

Step 1: Taking into account the different measurements of each indicator in each capacity, first standardize the indicators:(1)$$Ind_{lq}^{\prime }=\frac{Ind_{lq}-Ind_{\min}}{Ind_{\max}-Ind_{\min}}$$
where $Ind_{lq}$ is the value of the indicator, $Ind_{lq}^{\prime }$ is the standardized value of indicator q in province l, $Ind_{\max}$ and $Ind_{\min}$ represent the maximum and minimum values of indicator q in all provinces, respectively. Where l = 1, 2, …, n; q = 1, 2, …, m.

Step 2: Calculate the weight of indicator q in province l.
(2)$$y_{lq}=Ind_{lq}^{\prime }/\sum_{l=1}^{n}Ind_{lq}^{\prime }$$
where $y_{lq}$ is the calculated share of indicator q in province l.

Step 3: Calculate the entropy value of indicator q.
(3)$$e_{q}=-\frac{1}{\ln(n)}\sum_{l=1}^{n}y_{lq}\ln\left(y_{lq}\right)$$
where $e_{q}$ is the calculated entropy value of indicator q.

Step 4: Calculate the coefficient of variation of indicator q.
(4)$$g_{q}=1-e_{q}$$
where $g_{q}$ is the calculated coefficient of variation of indicator q.

Step 5: Calculate the weight of indicator q in all indicators.
(5)$$z_{q}=g_{q}/\sum_{q=1}^{m}g_{q}$$
where $z_{q}$ is the calculated weight of indicator q in all indicators.

Step 6: Calculate the composite score of each dimension for each province.
(6)$$c_{l}=\sum_{q=1}^{m}z_{q}X_{lq}^{\prime }$$
where $c_{l}$ is the calculated composite score of each dimension for each province.

After determining the composite scores of each dimension for each province, the composite scores of each dimension were summed to obtain the composite score of LRRR in China by referring to Peng et al. [51].

### 3.3. Dagum Gini Coefficient and Decomposition Method

This study focused on differences between regions and their characteristics, based on the measured LRRR, to provide a basis for narrowing the gap in LRRR in China and achieving coordinated development of rural regions. Therefore, this study measured the inter-regional differences in LRRR in China with the help of the Gini coefficient method, which is commonly used to measure inequality. The Dagum Gini coefficient, which was originally used to measure income inequality [54], can effectively address the issue of the sources of spatial differences and the overlap in groups [53,55] and has been widely used in economic, environmental, and energy studies [56,57,58]. To examine spatial differences in LRRR in China, and their sources, this study used the Dagum Gini coefficient for calculation and decomposition. The Dagum Gini coefficient G was calculated as shown in Equation (7).
(7)$$G=\frac{\sum_{j=1}^{k}\sum_{h=1}^{k}\sum_{i=1}^{n_{j}}\sum_{r=1}^{n_{h}}\left|LRRR_{ji}-LRRR_{hr}\right|}{2n^{2}\bar{LRRR}}$$
where n is the number of all provinces, k is the number of regions, j and h are subscripts for different regions, and i and r are subscripts for provinces. $n_{j}$ ($n_{h}$) is the number of provinces within the j (h) region. $LRRR_{ji}$($LRRR_{hr}$) is the livelihood resilience of i (r) for provinces within the j (h) region, and $\bar{LRRR}$ is the average LRRR in all provinces.

After measuring the differences in LRRR in China, this study further decomposed the sources of differences in terms of cluster decomposition using the decomposition method of Dagum Gini coefficient. The clusters were based on regional groups in China, including Eastern, Central, and Western Regions. The overall Gini coefficient G was decomposed into three components: intra-regional differences $G_{w}$ within the Eastern, Central and Western Regions, inter-regional differences $G_{nb}$ between the Eastern, Central and Western Regions, and the hypervariable density $G_{t}$ between the Eastern, Central and Western regions. The relationship between the three satisfies: $G=G_{w}+G_{nb}+G_{t}$; where the Gini coefficient $G_{jj}$ of region j and the intra-provincial differences $G_{w}$ within the region are calculated in Equations (8) and (9), respectively; the Gini coefficient $G_{jh}$ between the regions j and h and inter-regional differences $G_{nb}$ are calculated in Equations (10) and (11), respectively; $G_{t}$ is calculated in Equation (12).
(8)$$G_{jj}=\frac{\sum_{i=1}^{n_{j}}\sum_{r=1}^{n_{j}}\left|LRRR_{ji}-LRRR_{jr}\right|}{2n_{j}^{2}\bar{LRRR_{j}}}$$
(9)$$G_{w}=\sum_{j=1}^{k}G_{jj}p_{j}s_{j}$$
(10)$$G_{jh}=\sum_{i=1}^{n_{j}}\sum_{r=1}^{n_{h}}\frac{\left|LRRR_{ji}-LRRR_{hr}\right|}{n_{j}n_{h}\left(\bar{LRRR_{j}}+\bar{LRRR_{h}}\right)}$$
(11)$$G_{nb}=\sum_{j=2}^{k}\sum_{h=1}^{j-1}G_{jh}\left(p_{j}s_{h}+p_{h}s_{j}\right)D_{jh}$$
(12)$$G_{t}=\sum_{j=1}^{k}\sum_{h=1}^{j-1}G_{jh}\left(p_{j}s_{h}+p_{h}s_{j}\right)\left(1-D_{jh}\right)$$

In Equation (9), $p_{j}=n_{j}/n$ and $s_{j}=n_{j}\bar{LRRR_{j}}/nLRRR$. In Equations (11) and (12), $p_{h}=n_{h}/n$ and $s_{h}=n_{h}\bar{LRRR_{h}}/nLRRR$. $D_{jh}$ is the relative impact of the LRRR between the regions j and h, and is calculated as shown in the following equations:(13)$$D_{jh}=\frac{d_{jh}-p_{jh}}{d_{jh}+p_{jh}}$$
(14)$$d_{jh}=\int_{0}^{\infty }dF_{j}\left(LRRR\right)\int_{0}^{y}\left(LRRR-x\right)dF_{h}\left(x\right)$$
(15)$$p_{jh}=\int_{0}^{\infty }dF_{h}\left(LRRR\right)\int_{0}^{y}\left(LRRR-x\right)dF_{j}\left(x\right)$$
where $F_{j}$($F_{h}$) denotes the cumulative distribution function of LRRR in the adjusted areas j (h). $d_{jh}$ is the difference in LRRR between area j and area h, and represents the weighted average of all $LRRR_{ji}-LRRR_{hr}>0$ in area j and area h, which is calculated in Equation (14); $p_{jh}$ represents the weighted average of all $LRRR_{hr}-LRRR_{ji}>0$ in area j and area h, which is calculated in Equation (15).

In the analysis reported in this paper, a larger value of the Dagum Gini coefficient within a region indicates larger intra-regional differences in LRRR. A larger value of the Dagum Gini coefficient between regions indicates more uneven development of LRRR between two regions. In addition, a larger contribution of the decomposed Gini coefficient indicates that it is an important source of overall imbalance.

### 3.4. Kernel Density Estimation

Measuring regional differences in LRRR in China provides insight into the pattern of inter-regional imbalances, while portraying the dynamic distribution helps to further clarify the spatial dynamic distribution of the differences in LRRR. In the study of dynamic distribution, kernel density estimation is a common nonparametric estimation method. Kernel density estimation can characterize the distribution of variables and identify the evolution pattern of absolute differences through intuitive and continuous density curves, which is a good complement to the Gini coefficient method [53]. Specifically, the kernel density curve of LRRR within region j is generated by the following function.
(16)$$f\left(x\right)=\frac{1}{n_{j}B}\sum_{i=1}^{n_{j}}K\left(\frac{LRRR_{ji}-\bar{LRRR}}{B}\right)$$
where B is the bandwidth, K(x) is the kernel density function, and the rest of the variables are consistent with the previous section. In general, the choice of different kernel density functions has little effect on the estimation results [55]. Therefore, referring to the study of Cui and Chang [59], this paper developed the analysis based on the Gaussian kernel function. The Gaussian kernel function is shown in Equation (17).
(17)$$K(x)=\frac{1}{\sqrt{2\pi }}\exp(-\frac{x^{2}}{2})$$

Based on the results of kernel density estimation, the characteristics of changes in LRRR can be observed in the curve images. The horizontal position of the kernel density curve of a single period can represent high or low LRRR. The height and width of the curve peaks can reflect the degree of aggregation of LRRR within the region. The number of peaks can portray the degree of polarization of the LRRR, and the distribution extension, i.e., the degree of trailing, can describe the distance between the provinces with the highest or lowest LRRR and other provinces, where more severe trailing represents higher intra-regional variation.

### 3.5. Convergence Models

The trend towards convergence of gaps between regions in LRRR indicates whether or not there is a catch-up momentum in LRRR and clarifies the inter-regional inequality in LRRR in China. Convergence refers to whether there is a tendency for differences in LRRR in China to gradually reduce over time. Referring to Rezitis [60], this study employed the most commonly applied methods of σ convergence, absolute β convergence, and conditional β convergence for the analysis. σ convergence refers to the trend of decreasing deviation of LRRR in each region over time, and this study used the coefficient of variation (CV) to measure the σ convergence [60]. The CV for province i within region j is calculated as follows.
(18)$$CV=\frac{\sqrt{{\sum_{i=1}^{n_{j}}\left(LRRR_{ij}-\bar{LRRR_{ij}}\right)}^{2}/n_{j}}}{\bar{LRRR_{ij}}}$$
where $LRRR_{ij}$, $\bar{LRRR_{ij}}$ and $n_{j}$ are consistent with the previous settings. If the value of CV at the end of the study period is larger than that at the beginning, then there is σ convergence in LRRR in China and vice versa.

β convergence means that as time progresses, areas with lower LRRR have higher increases to catch up with areas with higher LRRR, and the gap between them gradually decreases and eventually reaches the same steady-state level [61]. β convergence can be further divided into absolute β convergence and conditional β convergence. Absolute β convergence refers to the tendency for LRRR to converge between regions without considering a set of factors that have a significant impact on LRRR [62]. The absolute β convergence model is as follows.
(19)$$\ln\left(\frac{LRRR_{i,t+1}}{LRRR_{it}}\right)=\alpha +\beta \ln\left(LRRR_{it}\right)+\mu_{i}+\eta_{t}+\varepsilon_{it}$$
where $LRRR_{i,t+1}$ denotes the LRRR in province i in period t+1, and $LRRR_{it}$ denotes the LRRR in province i in period t. The individual effects, time effects, and random disturbance terms are denoted by $\mu_{i}$, $\eta_{t}$, and $\varepsilon_{it}$, respectively. β is the convergence coefficient, if β<0 indicates that the LRRR in the region has a convergence trend, and the rate of convergence is v=−ln(1−|β|)/T.

This study considered the possible spatial correlations in LRRR arising from the acceleration of resource flows and the enhancement of inter-regional interaction effects [63]. Based on this, a neighboring spatial weight matrix was constructed and a dynamic spatial panel model was applied to identify the β convergence characteristics of LRRR in three major regions of China. Common spatial econometric models include the spatial autoregressive model (SAR), the spatial error model (SEM), the spatial Durbin model (SDM), etc. [64]. However, since the spatial correlation in LRRR in China is not known, this study fitted each of the above three models to select a suitable model for convergence analysis.

Whether spatial correlation exists and what form of spatial econometric model is chosen generally follows the following rules: First, a general panel regression model is constructed and a spatial autocorrelation is tested using the LM statistic; if spatial autocorrelation exists, this indicates that at least one of the SAR and SEM models holds. Next, the SDM model is constructed and the Wald statistic and LR statistic are used to determine whether it can be simplified to a SAR model or a SEM model [65]. After passing a series of tests, the main spatial econometric models in this paper include the spatial autocorrelation model (SAR), the spatial error model (SEM), and the spatial Durbin model (SDM), whose specific econometric models with absolute spatial β convergence are shown in Equations (20)–(22), respectively.
(20)$$\ln\left(\frac{LRRR_{i,t+1}}{LRRR_{it}}\right)=\alpha +\beta \ln\left(LRRR_{it}\right)+\rho \sum_{j=1}^{n}w_{ij}\ln\left(\frac{LRRR_{i,t+1}}{LRRR_{it}}\right)+\mu_{i}+\eta_{t}+\varepsilon_{it}$$
(21)$$\ln\left(\frac{LRRR_{i,t+1}}{LRRR_{it}}\right)=\alpha +\beta \ln\left(LRRR_{it}\right)+\mu_{i}+\eta_{t}+u_{it}\;u_{it}=\lambda \sum_{j=1}^{n}w_{ij}u_{it}+\varepsilon_{it}$$
(22)$$\begin{array}{ll} \ln\left(\frac{LRRR_{i,t+1}}{LRRR_{it}}\right)= & \alpha +\beta \ln\left(LRRR_{it}\right)+\rho \sum_{j=1}^{n}w_{ij}\ln\left(\frac{LRRR_{i,t+1}}{LRRR_{it}}\right)\\ & +\gamma \sum_{j=1}^{n}w_{ij}\ln\left(LRRR_{it}\right)+\mu_{i}+\eta_{t}+\varepsilon_{it} \end{array}$$
where ρ is the spatial lag coefficient which indicates the effect of the growth rate of LRRR in neighboring provinces, λ is the spatial error coefficient which indicates the spatial effect present in the random disturbance term, and γ is the spatial lag coefficient of the independent variable which indicates the effect of the LRRR in neighboring provinces. The $w_{ij}$ are the spatial neighboring weights, and fixed effects are determined by the Hausman test. The remainder of the variables are set consistent with Equation (18).

Considering the realistic situation, the conditional β convergence model adds a series of control variables to the absolute β convergence model, and the ordinary panel, SAR, SEM, and SDM models under the conditional β convergence are shown in Equations (23)–(26), respectively.
(23)$$\ln\left(\frac{LRRR_{i,t+1}}{LRRR_{it}}\right)=\alpha +\beta \ln\left(LRRR_{it}\right)+\delta X_{i,t+1}+\mu_{i}+\eta_{t}+\varepsilon_{it}$$
(24)$$\begin{array}{ll} \ln\left(\frac{LRRR_{i,t+1}}{LRRR_{it}}\right)= & \alpha +\beta \ln\left(LRRR_{it}\right)+\rho \sum_{j=1}^{n}w_{ij}\ln\left(\frac{LRRR_{i,t+1}}{LRRR_{it}}\right)\\ & +\delta X_{i,t+1}+\mu_{i}+\eta_{t}+\varepsilon_{it} \end{array}$$
(25)$$\ln\left(\frac{LRRR_{i,t+1}}{LRRR_{it}}\right)=\alpha +\beta \ln\left(LRRR_{it}\right)+\delta X_{i,t+1}+\mu_{i}+\eta_{t}+u_{it}\;u_{it}=\lambda \sum_{j=1}^{n}w_{ij}u_{it}+\varepsilon_{it}$$
(26)$$\begin{array}{ll} \ln\left(\frac{LRRR_{i,t+1}}{LRRR_{it}}\right)= & \alpha +\beta \ln\left(LRRR_{it}\right)+\rho \sum_{j=1}^{n}w_{ij}\ln\left(\frac{LRRR_{i,t+1}}{LRRR_{it}}\right)\\ & +\gamma \sum_{j=1}^{n}w_{ij}\ln\left(LRRR_{it}\right)+\delta X_{i,t+1}+\mu_{i}+\eta_{t}+\varepsilon_{it} \end{array}$$
where $X_{i,t+1}$ is a set of control variables affecting LRRR, and δ is a parameter vector. Referring to the established studies on the factors influencing LRRR, the control variables in the conditional β convergence model selected in this study include the economic level, urbanization rate, industrial structure, and the degree of marketization.

The economic level was measured using GDP per capita. The higher the level of economic development, the greater the ability of local rural residents to cope with adverse changes in the external environment [30]. The urbanization rate was measured as the proportion of the urban population relative to the total population of the province. The higher the urbanization rate, the more people there will be in urban areas, and the expansion of urban areas inevitably squeezes the living space of rural residents [5]. The industrial structure was measured by the share of GDP accounted for by the output value of the primary industry of each province. The development of the primary industry can significantly increase the income of rural residents [50]. The degree of marketization was measured by the marketization index. The marketization process not only affects rural residents’ agricultural investment and population mobility but also affects the overall rural consumption level [50].

### 3.6. Data

To maximize data availability and consistency, the research sample was panel data from 30 Chinese provinces (excluding Tibet) from 2006 to 2020. Data on livestock rearing, possession of agricultural machinery, garden area, area of crops cultivated, per capita net income of rural residents, postal delivery routes, per capita GDP, urbanization rate, and the share of the output value of the primary industry in each province were obtained from the China Statistical Yearbook [66]. Data on agricultural fixed asset investment, fiscal expenditure on agriculture, fiscal expenditure on minimum living allowance, rural doctors, education expenditure, and the number of agricultural technical training graduates were obtained from the China Rural Statistical Yearbook [67]. The marketization index was obtained from the China Market Index Database [68]. Descriptive statistics of the control variables in the spatial panel regressions are shown in Table 2.

## 4. Results and Discussion

### 4.1. Evaluation of the Livelihood Resilience of Rural Residents

Table 3 reports the composite LRRR scores for 30 provinces of China for each five years from 2006 to 2020. Figure 1 illustrates the livelihood resilience of 30 provinces in three regions in 2006 and 2020. Figure 2 shows the mean values of the composite LRRR scores overall China and in the Eastern, Central, and Western Regions during the examination period, and their changes.

Overall, the average livelihood resilience score of rural residents in China from 2006 to 2020 gradually decreased from 0.764 to 0.739, with an average annual decrease of 0.237%. This indicates that the ability of Chinese rural residents to cope with changes in the external environment weakened, although there was some fluctuation. By periods, the average livelihood resilience scores of Chinese rural residents increased in 2008–2010, 2012–2013, 2015–2016, and 2018–2019, with a maximum increase of 6.583%, while they decreased during the remainder of the periods, with a maximum decrease of 5.305%.

From a regional perspective, the average livelihood resilience scores of rural residents in the Eastern and Central Regions had different degrees of decline (See Figure 2), from 1.055 and 0.586, respectively, to 0.923 and 0.504, with annual average decreases of 0.950% and 1.071%, with the Eastern Region remaining above the national average during the examined period. The average LRRR in the Western Region rose from 0.603 to 0.726 and reached the level of the national average at the end of the study period, with an average annual increase of 1.335%. The average livelihood resilience scores of rural residents in the three major regions were increasing in 2007–2010, 2012–2013, and 2015–2016, with the largest increases of 11.223%, 9.091%, and 9.480% in the Eastern, Central, and Western Regions, respectively, while the declines were concentrated in 2011–2012, 2013–2014, and 2016–2018, with maximum decreases of 6.862%, 3.090%, and 9.121%.

The LRRR in the Eastern Region was significantly higher than those in the Central and Western Regions during the study period (See Figure 2). This corroborates with the conclusion of a previous study that the livelihood resilience of pastoralists in China was high in the Eastern Region and low in the Western Region [69]. However, the trends in development of the LRRR varied among regions. The LRRR in the Western Region increased significantly; for example, the LRRR in Yunnan, Guizhou, and Qinghai increased by more than 50%. In contrast, the LRRR in the Eastern and Central Regions decreased; for example, the LRRR in Liaoning, Shanxi, and Jilin decreased by more than 30%. In addition, the difference in the LRRR between Eastern and Western Regions gradually decreased over time, while the difference between the Central and Western Regions gradually increased. This shows that there were large regional differences in LRRR in China. To investigate the underlying causes of regional differences in LRRR in China, it is necessary to further measure the differences in LRRR across regions and to analyze their specific sources.

### 4.2. Decomposition of Regional Differences in the Livelihood Resilience of Rural Residents

#### 4.2.1. Overall and Intra-Regional Differences of the Livelihood Resilience of Rural Residents

The evolution of the overall and within-region Gini coefficients of LRRR in China is shown in Figure 3. At the national level, from the beginning to the end of the period, the difference in LRRR in China showed a small decrease. The overall Gini coefficient had a mean value of 0.258 during the study period and fluctuated, reaching a minimum value of 0.229 and a maximum value of 0.280 in 2016 and 2019, respectively, which indicates that the LRRR within China had obvious variations between provinces.

In intra-regional comparisons, the Gini coefficients of the Central and Western Regions did not exceed the overall Gini coefficient (See Figure 3), indicating that imbalance between provinces within the Central and Western Regions was relatively low. The average value of the Gini coefficient in the Eastern Region reached 0.263 and grew faster after 2013, reaching a peak of 0.290 in 2019. The average value of the Gini coefficient in the Central Region was 0.152 and remained low after 2013, reaching a minimum value of 0.105 in 2020. This indicates that the LRRRs of the provinces within the Central Region were very similar. This may be due to the well-developed agriculture in the central provinces and the small differences in economic development, which can rapidly reduce the gap in LRRR in line with the “Rise of Central China” strategy [70]. The mean value of the Gini coefficient in the Western Region was 0.192, which indicates that the gap in LRRR was relatively small among the provinces within the region. The Gini coefficient peaked at 0.214 in 2010, but then the intra-regional gap narrowed and the LRRR in all provinces converged to the same level.

#### 4.2.2. Inter-Regional Differences of the Livelihood Resilience of Rural Residents

The inter-regional differences in LRRR of China as measured by the Dagum Gini coefficient are shown in Figure 4.

Considering the variation in LRRR between regions, the variation between the Central and Western Regions was smallest, with a sample mean of 0.189 and minimum and maximum values of 0.169 and 0.209, respectively. The variations between the Eastern and Central Regions and the Eastern and the Western Regions were larger, with sample means of 0.275 and 0.262, respectively. The minimum and maximum values of the difference between the Eastern and Central Regions were 0.305 and 0.250, respectively, and the minimum and maximum values of the difference between the Eastern and Western Regions were 0.232 and 0.293, respectively. 

The differences between the Eastern and Central Regions, and the Central and Western Regions were increasing, and the differences between the Eastern and Western Regions were significantly decreasing (See Figure 4). The difference between the Central and Western Regions increased the most significantly, with an average annual increase of 0.687%, but these were still the two regions with the smallest inter-regional differences. The difference between the Eastern and Central Regions in LRRR increased slightly, with an average annual increase of 0.130%. The differences between the Eastern and Western Region declined more significantly, with an average annual decrease of 0.411%, but the degree of inter-regional differences remained at a high level. This again indicates that the LRRR in Central and Western China lagged behind that of the Eastern Region, with relatively flat changes in the differences between the Eastern and Central Regions, and a larger range of fluctuations in the differences between the Eastern and Western Regions and the Central and Western Regions. This indicates that the Eastern and Central Regions were developing LRRR at a more consistent pace, while there is more pressure to narrow the gap between the Eastern and Central Regions and the Eastern and Western Regions.

#### 4.2.3. Spatial Differences and Decomposition of the Livelihood Resilience of Rural Residents

The overall differences in LRRR can be decomposed into three components: intra-regional, inter-regional, and hypervariable density, using the method described previously. Figure 5 shows the value of each component’s contribution to the overall Gini coefficient, and the percentage contributed.

The contribution of intra-regional variation to the Gini coefficient was 0.074 at the beginning of the study period (See Figure 5); it then increased, and then decreased with 2013 as the inflection point, finally rising to 0.076 at the end of the study period, with an average contribution of 0.077 during the period. The contribution rate of intra-regional differences also increased, with 2016 as the inflection point, and then decreased, fluctuating from 28.654% at the beginning to 29.995% at the end of the study period, with an average contribution of 25.954% during the period.

The contribution of inter-regional differences to the Gini coefficient of the LRRR generally showed a downward trend (See Figure 5). The initial value was 0.142, and it then decreased to 0.121 at the end of the study period; the average contribution value during the investigation period was 0.127. The contribution rate of inter-regional differences also decreased from 54.849% at the beginning to 47.826% at the end of the study period, with an average contribution of 49.329%. The inter-regional difference in LRRR was the most important cause of the overall difference, which is consistent with the previous findings in this study, and suggests that reducing the difference between regions should be the focus of promoting the development of LRRR in China in future.

The contribution of the hypervariable density to the Gini coefficient and its contribution rate significantly increased during the study period (See Figure 5). The contribution values were 0.043 at the beginning and 0.056 at the end of the study period, with an average of 0.053. The contribution rate was 16.497% at the beginning and it rose to 22.205% at the end, with an average contribution rate of 20.719%. The contribution of the hypervariable density reflects the degree of overlapping among Eastern, Central, and Western Regions. It can be seen that the interaction between intra-regional differences and inter-regional differences gradually increased its impact upon the overall differences in LRRR over time.

### 4.3. Dynamic Distribution of the Livelihood Resilience of Rural Residents

The Dagum Gini coefficient revealed the numerical levels of the differences in LRRR in China overall, together with their specific sources, and identified the differences among regions, but was unable to describe the dynamic evolution of the distribution. This study used kernel density estimation to characterize the distribution of livelihood resilience in each region, focusing on key attributes such as the distribution of the density curve, the center position of the distribution curve, the extension of the main peak, and the number of peaks. The corresponding characteristics of dynamic evolution are reported in Table 4, and the specific kernel density estimation results are shown in Figure 6.

Considering the location of the distribution curve, the center of the China-overall curve shows a leftward shift (See Figure 6). This indicates that LRRR declined in most provinces over time. The center position of the distribution curve in the Eastern Region shows a small leftward shift, and there was a small decline in LRRR within the Eastern Region over time. The centers of the distribution curves in the Central and Western Regions show a rightward shift during the period, suggesting that the LRRRs in the Central and Western regions have increased, which is generally consistent with the factual characteristics described in the previous section.

Considering the shape of the kernel density curves, the height of the main peak first decreased, then increased, and the width decreased (See Figure 6). This indicates that the overall dispersion of LRRR in China tended to decrease. The heights of the peaks in the Eastern and Central Regions also decreased, followed by a general increase and a decreasing width, indicating that the absolute differences in LRRR between provinces within the Eastern and Central Regions were decreasing. In the Western Region, the height of the main peak of the distribution curve decreased and the width increased, implying that the dispersion of rural residents’ livelihood resiliences between provinces in the Western Region was increasing and the absolute difference between provinces widened.

Considering the extension of the main peak, there is a significant right-hand tail in the distribution curves for China-overall and three regions. This indicates that the resilience of livelihood of rural residents in some provinces within each region was significantly higher than those of other provinces within the same region; but the extensions of the distribution curves differ. The extension in China-overall and the Eastern Region show a widening trend, which means that the intra-regional gap in China-overall and the Eastern Region continued to widen. This can be explained by provinces such as Beijing, Zhejiang, and Jiangsu having much higher livelihood resiliencies than other provinces in the region. The kernel density distribution curves of LRRR in the Central and Western Regions show a converging trend, which indicates that the intra-regional gap in the Central and the Western Region further narrowed over time.

Considering the number of peaks, the distribution curves for China-overall and the three regions consist of double or multiple peaks (See Figure 6). This indicates that there was some polarization or even multi-polarization in LRRR within the regions. Specifically, the peaks in China-overall develop from double peaks to multiple peaks, and the emergence of peaks over time indicates a weak gradient effect and a multipolar divergence in LRRR in each province. The Eastern Region also evolves from double peaks to multiple peaks, with relatively large distances between the main and side peaks, and there is a more pronounced spatial polarization, with obvious disparities in the LRRR in the eastern provinces. The result for the Central Region is dominated by single peaks, and although double peaks appear at the end of the period, the polarization of LRRR in each province in the region tends to be weaker in general, indicating that LRRR in the Central Region tends to be at a more uniform level. In the Western Region, the peaks evolve from multiple peaks to a single peak, and the polarization of LRRR in each province in the region moderated, indicating that the differences in LRRR among the provinces in the Western Region have narrowed and converged.

### 4.4. Convergence Analysis of the Livelihood Resilience of Rural Residents

#### 4.4.1. Convergence of the Livelihood Resilience of Rural Residents

The σ convergence of LRRR in each region of China are shown in Table 5. The CV of China-overall shows a repeated rise and fall, with the maximum occurring in 2007 and the minimum in 2016. The CV at the end of the period was larger than that of the beginning, so there is σ convergence. The CV of the Eastern Region had a pattern of rising-declining-rising, with the maximum appearing in 2019 and the minimum in 2013. The CV at end of the period was larger than that of the beginning of the period, so there was no σ convergence in general. The CV of the Central Region showed a slight increase and then a significant decrease, with the maximum in 2009 and the minimum in 2020. The CV at the end of the period was significantly smaller than the CV at the beginning of the period, so there is an obvious σ convergence trend. The CV of the Western Region showed a fluctuation of rise and decline, with the maximum appearing in 2014 and the minimum appearing at the beginning of the period in 2006. The CV at the end of the period was larger than the CV of the beginning of the period, so there is no σ convergence.

In general, the CVs of China-overall and the Eastern and Western Regions increased to different degrees; the regional gap in LRRR widened and the regional imbalance was highlighted. The CV of the Central Region decreased significantly, and the intra-regional difference in LRRR narrowed, which also confirms the results of the Gini coefficients in Section 4.3.

#### 4.4.2. Convergence of the Livelihood Resilience of Rural Residents

Table 6 shows the absolute β convergence of LRRR for each region of China. Since different regions may have different patterns of spatial effects on LRRR, the LM test was first applied to determine whether there was a spatial autocorrelation effect on the absolute β convergence of each region, then the Wald test, the LR test, and the Hausman test were used to select the optimal spatial model forms and fixed effects. The spatial Durbin model (SDM) was chosen for China-overall and the Central region, and the spatial error model (SEM) was chosen for the Eastern and Western Regions.

The regression results show that, first, there was absolute β convergence in LRRR in all regions of China. The convergence coefficients for overall China and the Eastern, Central, and Western Regions were all significant at the 1% level, indicating that LRRR converged to respective steady-state levels in the long run when the effects of a range of economic and social factors on LRRR were not considered.

Second, there were differences in the rate of convergence of LRRR across regions. The convergence rates were 2.6%, 1.8%, 1.9%, and 3% for China-overall and the Eastern, Central, and Western Regions, respectively (See Table 6).

Third, China-overall and all the regions exhibited different spatial effects. Spatial lags of independent and dependent variables existed in China-overall and the Western Region, and both ρ and γ for each model were positively significant at least at the 5% level of significance, indicating that the rates of change in LRRR in the provinces within the regions were simultaneously affected by positive spatial spillovers from the rates of change in other provinces. Spatial error lags existed in the Eastern and Central Regions, and the λ in the models for both regions were significant at the 1% level, indicating that the error terms in the spatial regressions of LRRR in one province within the region were spatially correlated and the error terms had spatial effects on other provinces within the region. However, the above analysis on the absolute β convergence of LRRR across the regions was conducted under the assumption that the levels of economic development, industrial structure, urbanization rate, and marketization degree were similar across regions. However, this is not in reality the case, and therefore further conditional β convergence studies are needed.

Table 7 shows the conditional β convergence results for LRRR in each region of China. After tests, the spatial Durbin model (SDM) was chosen for overall China, and the spatial error model (SEM) was chosen for the Eastern, Central, and Western Regions. The results show that, first, there was conditional β convergence for LRRR in all regions. The CVs of China-overall and the Eastern, Central, and Western Regions were all significant at the 1% level, indicating that the trend towards convergence of LRRR still exists in all regions after considering economic and social factors. Second, compared with the absolute β convergence, the conditional β convergence rate accelerated in all regions except the Central Region, where the rate of convergence remained unchanged. The convergence rates were 3.3%, 2.3%, 1.9%, and 5.1% for China-overall and the Eastern, Central, and Western Regions, respectively (See Table 7). Third, China-overall and the three regions showed different spatial effects, but the spatial effects in individual regions differed from those of the absolute β convergence analysis. Among them, the type of spatial effect in the Central Region changed from SDM to SEM, indicating that the spatial spillover effect of provinces within the Central Region disappeared to some extent. Apart from this, it did not differ from the absolute β convergence analysis.

## 5. Conclusions and Policy Implication

### 5.1. Conclusions

Academic research has recently concentrated on ways to make LRRR sustainable. Improving LRRR is not only a key issue in China’s current rural construction and governance and the realization of the “Rural Revitalization” strategy. For many developing countries, improving LRRR is also of great value in advancing modernization. The research model developed in this work advances the dynamic comprehension of LRRR and offers suggestions for the optimization and calculation of the index system measuring the LRRR in developing nations. The research findings also offer developing nations policy guidance for achieving the sustainable livelihood resilience.

This study constructed a multidimensional livelihood resilience indicator system for rural residents and measured the LRRR in each province of China using the entropy method. On this basis, the regional differences, sources, and dynamic distribution of LRRR in China-overall and the Eastern Central and Western Regions were specifically analyzed using the Dagum Gini coefficient and kernel density estimation methods. The CV and spatial panel convergence models were employed to test the convergence characteristics in China as a whole and in the Eastern, Central, and Western Regions. The results showed that: first, the spatial distribution of LRRR in China was uneven across provinces, with an east to west spatial gradient towards lower livelihood resilience. The average value of LRRR in the Eastern Region was higher than the overall average, while those in the Central and Western Regions were lower than average. The LRRR in the Western Region had a fluctuating upward trend, while the Eastern and Central regions had declining trends.

Second, in terms of variation characteristics, the dispersion of LRRR in China-overall tended to narrow and polarize. In addition, the regional differences between provinces in China showed an oscillating upward trend, indicating that the unevenness of rural residents’ livelihood resiliences expanded nationwide. The differences within the Eastern and Western Regions had a widening trend, while the intra-regional differences in China-overall and in the Central Region had a narrowing trend. Regional differences between the Eastern and Central Regions and the Eastern and Western Regions were much higher than that between the Central and Western Region. The inter-regional differences were the main source of the overall differences within China, and the intra-regional differences were the second source, with the lowest contribution to the hypervariable density.

Third, in terms of convergence characteristics, the coefficients of σ convergence of LRRR in China-overall and the Central Region had a decreasing trend, and the differences among provinces within the region were narrowing. The coefficients of σ convergence for LRRR in the Eastern and Western Regions were increasing to some extent, and the intra-regional differences were widening. There were absolute β convergence and conditional β convergence for LRRR in overall China and in the Eastern, Central, and Western Regions. These results indicate that the changes in LRRR in China as a whole and in the Eastern, Central, and Western Regions eventually converged to the same steady-state level over time. The rate of conditional β convergence, after considering the effects of control variables, all increased to different degrees compared with that of the absolute β convergence, indicating that economic and social factors such as economic level, industrial structure, urbanization rate, and degree of marketization accelerated the convergence of differences to some extent.

### 5.2. Policy Implication

Based on the above findings, we draw the following policy implications: First, it is important to fully understand the reality that the LRRR in China is on the decline. Although the LRRR in the Western Region had increased, the overall LRRR in China was still declining, and there is much room for improvement. We should fully understand the important role of livelihood resilience in sustainable rural development, improve rural public service infrastructure, reduce the vulnerability of rural residents to external shocks, and enhance their adaptability. Second, the spatially uneven distribution of LRRR in China should be fully recognized. The difference in LRRR between provinces in China-overall and the Central Region has narrowed, while the difference in LRRR between provinces in the Eastern and Western Regions has increased. While promoting the steady improvement of rural residents’ livelihood resilience, we should continue to strengthen the awareness of regional spatial synergy and explore the reasons for the existence and expansion of regional differences. At the same time, with the goal of balanced development, equal development opportunities for rural residents in all regions, especially disadvantaged regions, should be guaranteed. The differences in livelihood resilience among rural residents in the Eastern, Central, and Western Regions was the main source of the differences in the LRRR in China as a whole. Therefore, when formulating relevant policies, policy preferences can be given to regions with lower livelihood resilience to reduce regional differences. Third, the convergence characteristics of rural residents’ livelihood resiliences suggest that, while reducing the inter-regional differences among the Eastern, Central, and Western Regions, it is important to pay attention to the coordination of the convergence rate of rural residents’ livelihood resiliences among regions. Meanwhile, provinces with higher LRRR should make use of the spatial spillover effect, amplify the radiation effect to neighboring areas, and simultaneously raise the livelihood resilience in neighboring provinces.

## Figures and Tables

**Figure 1 ijerph-19-10612-f001:**
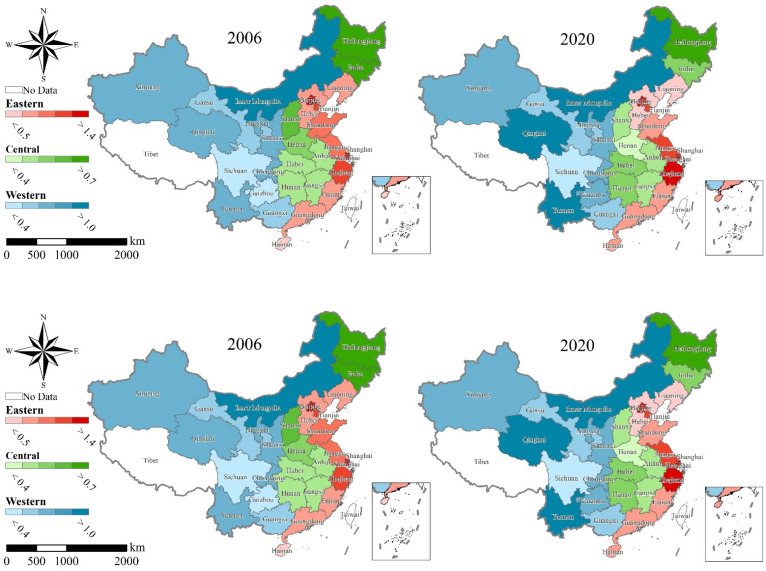
Livelihood resilience of rural residents in China by province in 2006 and 2020. Source: the authors’ own work.

**Figure 2 ijerph-19-10612-f002:**
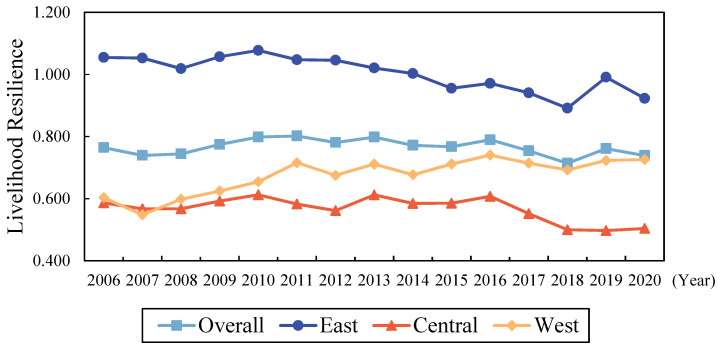
Year-to-year changes in livelihood resilience of rural residents by region in China. Source: the authors’ own work.

**Figure 3 ijerph-19-10612-f003:**
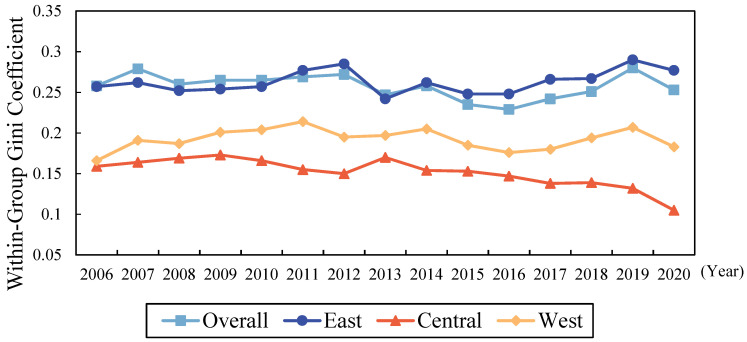
Variation of the overall and intra-regional differences of the livelihood resilience of rural residents in China. Source: the authors’ own work.

**Figure 4 ijerph-19-10612-f004:**
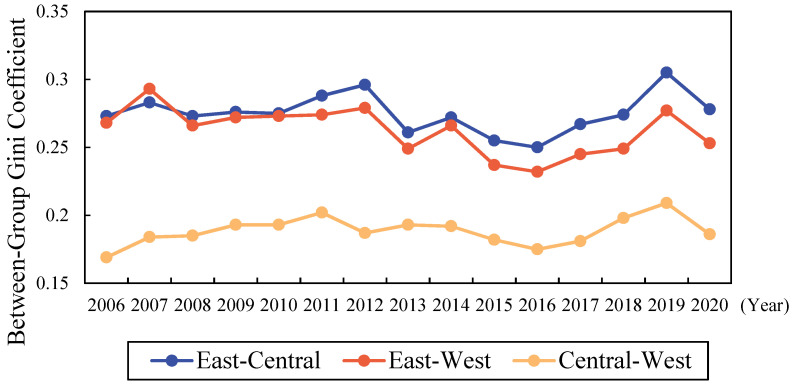
Variation of the Inter-regional Differences in the Livelihood Resilience of Rural Residents in China. Source: the authors’ own work.

**Figure 5 ijerph-19-10612-f005:**
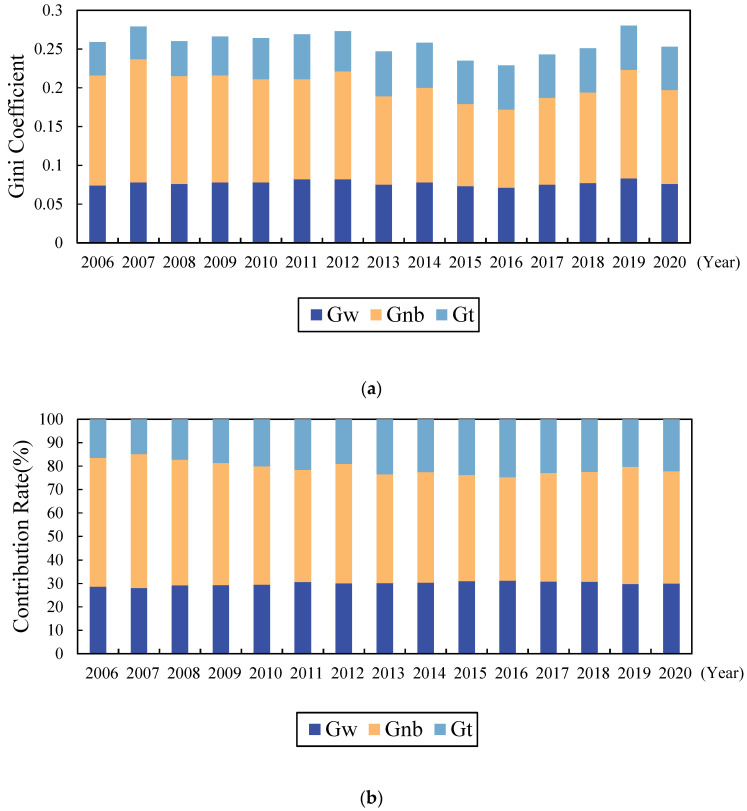
Spatial differences and sources in the livelihood resilience of rural residents in China: (**a**) contribution value; (**b**) contribution rate. Source: the authors’ own work.

**Figure 6 ijerph-19-10612-f006:**
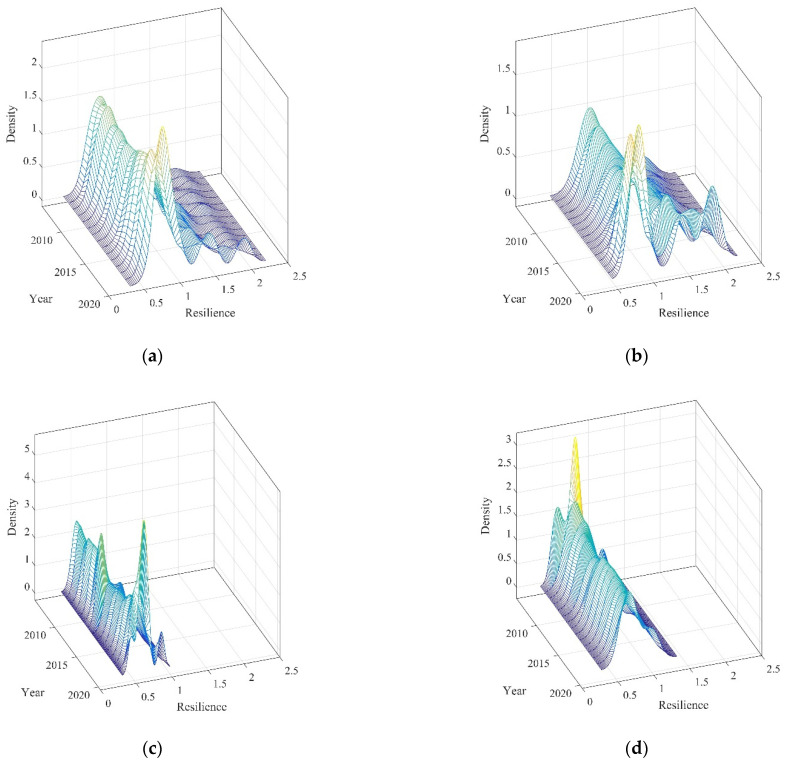
Dynamic distribution of the livelihood resilience of rural residents in three regions of China, and overall: (**a**) China-overall; (**b**) Eastern; (**c**) Central; (**d**) Western. Note: The lighter the color, the greater the density. Source: the authors’ own work.

**Table 1 ijerph-19-10612-t001:** Definitions and units of measurement of the indicators of the livelihood resilience of rural residents.

Livelihood Capacities	Indicators	Description and Definition
Buffer capacity	Livestock rearing	Livestock rearing per rural resident (head/person)
	Possession of agricultural machinery	Total power of agricultural machinery per capita (kW/person)
	Garden area	Area of orchards and tea plantations per rural resident (ha/person)
	Crop cultivated area	Area of major crops cultivated per capita (ha/person)
	Per capita income	Rural per capita net income (10,000 yuan/person)
	Agricultural fixed asset investment	Agricultural fixed asset investment per rural resident (10,000 yuan/person)
Self-organization capacity	Fiscal expenditure on agriculture	Fiscal expenditure on agriculture, forestry, and water affairs (million yuan/person)
	Fiscal expenditure on minimum living allowance	Rural residents’ per capita minimum living allowance (million yuan/person)
	Medical care	Number of rural doctors per 1000 agricultural population (persons/1000)
	Postal delivery routes	Postal delivery routes per 1000 agricultural population (km/1000 people)
Learning capacity	Education expenditure	Per capita education expenditure of rural residents (yuan/person)
	Percentage of agricultural skills training	Number of agricultural technical training graduates/number of the rural population (%)

Source: the authors’ own work.

**Table 2 ijerph-19-10612-t002:** Descriptive statistics of variables.

Variables	Description	Obs.	Mean	SD	Min	Max
lngdp	The logarithm form of GDP divided by population	420	10.489	0.600	8.717	11.994
urban	Urban population in the total population of the province	420	55.100	13.760	27.460	89.600
industrial	The ratio of the output value of the first industry to GDP	420	10.330	5.453	0.300	32.700
market	Marketization index of each province	420	6.273	1.763	2.330	11.710

Source: the authors’ own work.

**Table 3 ijerph-19-10612-t003:** Livelihood resilience of rural residents in China by province.

Region	Province	2006	2010	2015	2020
Eastern	Beijing	2.137	2.039	1.955	1.627
	Tianjin	1.289	1.306	1.357	1.125
	Hebei	0.726	0.614	0.514	0.458
	Liaoning	0.800	0.857	0.821	0.458
	Shanghai	1.865	1.860	1.244	1.525
	Jiangsu	0.895	1.322	1.204	1.198
	Zhejiang	1.291	1.316	1.108	1.611
	Fujian	0.717	0.677	0.584	0.560
	Shandong	0.824	0.770	0.654	0.502
	Guangdong	0.594	0.470	0.537	0.572
	Hainan	0.464	0.621	0.531	0.515
	Eastern Average	1.055	1.077	0.955	0.923
Central	Shanxi	0.692	0.677	0.638	0.470
	Jilin	0.811	0.810	0.714	0.542
	Heilongjiang	0.880	1.003	0.915	0.722
	Anhui	0.410	0.494	0.397	0.441
	Jiangxi	0.428	0.467	0.421	0.478
	Henan	0.525	0.538	0.452	0.337
	Hubei	0.492	0.509	0.619	0.521
	Hunan	0.453	0.401	0.525	0.519
	Central Average	0.586	0.612	0.607	0.504
Western	Inner Mongolia	1.054	1.190	1.228	1.058
	Guangxi	0.431	0.398	0.396	0.470
	Chongqing	0.609	0.574	0.659	0.606
	Sichuan	0.370	0.394	0.404	0.449
	Guizhou	0.342	0.372	0.539	0.638
	Yunnan	0.637	0.643	0.825	1.026
	Shaanxi	0.652	0.757	0.752	0.616
	Gansu	0.486	0.468	0.583	0.486
	Qinghai	0.683	0.924	0.981	1.182
	Ningxia	0.666	0.687	0.630	0.688
	Xinjiang	0.704	0.792	0.828	0.765
	Western Average	0.603	0.654	0.711	0.726
Overall Average		0.764	0.798	0.767	0.739

Source: the authors’ own calculations.

**Table 4 ijerph-19-10612-t004:** Characteristics of the dynamic distribution of the livelihood resilience of rural residents in China.

Region	Distribution Location	Shape of Curves	Extension of the Main Peak	Number of Peaks
China-overall	Shift left	Increase in height and decrease in width	Right trailing and extension widened	Double or multiple peaks
Eastern	Shift left	Increase in height and decrease in width	Right trailing and extension converged	Double or multiple peaks
Central	Shift Right	Increase in height and decrease in width	Right trailing and extension widened	Single or double peaks
Western	Shift Right	Decrease in height and increase in width	Right trailing and extension converged	Single or multiple peaks

Source: the authors’ own work.

**Table 5 ijerph-19-10612-t005:** Convergence of the livelihood resilience of rural residents in China by region.

Year	Overall	Eastern	Central	Western
2006	0.879	0.484	0.292	0.314
2007	0.936	0.485	0.303	0.360
2008	0.866	0.472	0.315	0.339
2009	0.868	0.471	0.322	0.362
2010	0.853	0.468	0.312	0.370
2011	0.859	0.501	0.286	0.383
2012	0.893	0.520	0.280	0.350
2013	0.770	0.438	0.306	0.371
2014	0.821	0.476	0.275	0.384
2015	0.750	0.460	0.279	0.333
2016	0.730	0.459	0.268	0.316
2017	0.786	0.497	0.254	0.319
2018	0.811	0.503	0.254	0.347
2019	0.901	0.536	0.244	0.371
2020	0.819	0.514	0.202	0.334

Souce: the authors’ own calculations.

**Table 6 ijerph-19-10612-t006:** Absolute β convergence of the livelihood resilience of rural residents in China by region.

Variables	(1)	(2)	(3)	(4)
	Overall SDM	Eastern SEM	Central SDM	Western SEM
β	−0.309 ***	−0.232 ***	−0.235 ***	−0.365 ***
	(0.037)	(0.057)	(0.070)	(0.065)
ρ or λ	0.348 ***	0.237 ***	0.351 ***	0.500 ***
	(0.058)	(0.078)	(0.079)	(0.072)
γ	0.328 ***		0.235 **	
	(0.058)		(0.094)	
Convergence rate	0.026	0.018	0.019	0.03
Spatial effect	YES	YES	YES	YES
Time Effect	YES	NO	NO	NO
Observations	420	154	112	154
Log-likelihood	515.727	173.744	135.563	170.532
R-squared	0.087	0.083	0.051	0.129

Notes: Standard errors are in parentheses. ***, ** denote significance at the 1% and 5% levels, respectively. Source: the authors’ own calculations.

**Table 7 ijerph-19-10612-t007:** Conditional β convergence of the livelihood resilience of rural residents in China by region.

Variables	(1)	(2)	(3)	(4)
	Overall SDM	Eastern SEM	Central SEM	Western SEM
β	−0.372 ***	−0.274 ***	−0.230 ***	−0.510 ***
	(0.039)	(0.057)	(0.068)	(0.071)
ρ or λ	0.283 ***	0.276 ***	0.329 ***	0.403 ***
	(0.061)	(0.087)	(0.084)	(0.081)
γ	0.183 **			
	(0.071)			
Control variables	Control	Control	Control	Control
Convergence rate	0.033	0.023	0.019	0.051
Spatial effect	YES	YES	YES	YES
Time Effect	YES	NO	NO	NO
Observations	420	420	112	154
Log likelihood	529.436	177.525	143.035	178.761
R-squared	0.002	0.110	0.103	0.293

Notes: Standard errors are in parentheses. ***, ** denote significance at the 1% and 5% levels, respectively. Source: the authors’ own calculations.

## Data Availability

Data within this paper are available from the corresponding author.

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
