# Peer review of "Spatio-Temporal Variation and Decomposition Analysis of Livelihood Resilience of Rural Residents in China"

_ijerph, 2022, doi:10.3390/ijerph191710612_

Round 1

Reviewer 1 Report

Following are my comments on the paper:  Spatio-temporal variation and decomposition analysis of liveli- 2 hood resilience of rural residents in China

This is a remarkable and very sophisticated paper which should have a major influence in China in improving the resilience of the rural population.

The writing is excellent and the English perfect. I could detect no faults at all.

Figure 5 The vertical axis does not have a legend. Should the dimensions be standardized? At present resilience axis varies from 2 (Overall & Eastern) to 1 – 1.4 (Central & Western). Making them the same would better enable comparisons between regions. These are excellent figures.

I suggest a map of China should be included showing the three regions – Eastern, Central and Western, and the provinces within each. The boundaries of each region should be clearly marked.

Some of the lengthy paragraphs could be broken up to make it easier for the reader.

The sections are: Introduction, materials and methods, results and discussion, conclusions and policy implications and each section is well covered. I detected no weaknesses, only strengths in the paper.

Author Response

      Thank you very much for giving us the precious opportunity to revise the manuscript. I had revised the manuscript in light of your comments after a careful discussion with the co-authors. The following are our detailed point-by-point answers to your queries. We hope the revised manuscript can comply with the high standard of papers accepted for publication.

Response to Reviewer #1

Following are my comments on the paper: Spatio-temporal variation and decomposition analysis of livelihood resilience of rural residents in China

This is a remarkable and very sophisticated paper which should have a major influence in China in improving the resilience of the rural population.

The writing is excellent and the English perfect. I could detect no faults at all.

Response: Thank you very much for your approval of our manuscript. The following are our detailed point-by-point answers to your queries.

  • Figure 5 The vertical axis does not have a legend. Should the dimensions be standardized? At present resilience axis varies from 2 (Overall & Eastern) to 1 – 1.4 (Central & Western). Making them the same would better enable comparisons between regions. These are excellent figures.

Response and revision: Thank you very much for your kindly and encouraging comment. We have added the legend of the vertical axis in the figures of kernel density estimation, and the dimensions of the resilience were standardized from 0 to 2.5 to facilitate the comparison of different regions. The optimized figure is shown in the attachment. Please see the attachment. 

  • I suggest a map of China should be included showing the three regions – Eastern, Central and Western, and the provinces within each. The boundaries of each region should be clearly marked.

Response and revision: Thank you very much for your helpful suggestion. We have added a map of China containing livelihood resilience of rural residents of 30 provinces in 2006 and 2020, and the boundaries of each region were marked through different colours. The figure is shown in the attachment. Please see the attachment.

  • Some of the lengthy paragraphs could be broken up to make it easier for the reader.

Response and revision: Thank you very much for your constructive comment. We have split some long paragraphs to make it easier to read as follows:

“…Considering the variation in LRRR between regions, the variation between the Central and Western Regions was smallest, with a sample mean of 0.189 and minimum and maximum values of 0.169 and 0.209, respectively. The variations between the Eastern and Central Regions and the Eastern and the Western Regions were larger, with sample means of 0.275 and 0.262, respectively. The minimum and maximum values of the difference between the Eastern and Central Regions were 0.305 and 0.250, respectively, and the minimum and maximum values of the difference between the Eastern and Western Regions were 0.232 and 0.293, respectively.

The differences between the Eastern and Central Regions, and the Central and Western Regions were on the rise, and the differences between the Eastern and West-ern Regions were significantly decreasing (See Figure 4.). The difference between the Central and Western Regions increased the most significantly, with an average annual increase of 0.687%, but these were still the two regions with the smallest inter-regional differences. The difference between the Eastern and Central Regions in LRRR increased slightly, with an average annual increase of 0.130%. The differences between the East-ern and Western Region declined more significantly, with an average annual decrease of 0.411%, but the degree of inter-regional differences remained at a high level. This again indicates that the LRRR in Central and Western China lagged behind that of the Eastern Region, with relatively flat changes in the differences between the Eastern and Central Regions, and a larger range of fluctuations in the differences between the Eastern and Western Regions, and Central and Western Regions. This indicates that the Eastern and Central Regions were developing LRRR at a more consistent pace, while there is more pressure to narrow the gap between the Eastern and Central, and Eastern and Western regions. …”

“…The  convergence of LRRR in each region of China are shown in Table 5. The CV of China overall shows a repeating rise and fall, with the maximum occurring in 2007 and the minimum in 2016. The CV at the end of the period was larger than that of the beginning, so there is  convergence. The CV of the Eastern Region had a pattern of rising-declining-rising, with the maximum appearing in 2019 and the minimum in 2013. The CV at end of the period was larger than that of the beginning of the period, so there was no  convergence in general. The CV of the Central Region showed a slight increase and then a significant decrease, with the maximum in 2009 and the minimum in 2020. The CV at the end of the period was significantly smaller than the CV at the beginning of the period, so there is an obvious  convergence trend. The CV of the Western Region showed a fluctuation of rise and decline, with the maximum appearing in 2014 and the minimum appearing at the beginning of the period in 2006. The CV at the end of the period was larger than the CV of the beginning of the period, so there is no  convergence.

In general, the CVs of China overall and the Eastern and Western Regions increased to different degrees; the regional gap in LRRR widened and the regional imbalance was highlighted. The CV of the Central Region decreased significantly, and the intra-regional difference in LRRR narrowed, which also confirms the results of the Gini coefficients in section 3.3. …”

“…Table 6 shows the absolute  convergence of LRRR for each region of China. Since different regions may have different patterns of spatial effects on LRRR, the LM test was first applied to determine whether there was a spatial autocorrelation effect on the absolute  convergence of each region, then the Wald test, LR test, and the Hausman test were used to select the optimal spatial model forms and fixed effects. The spatial Durbin model (SDM) was chosen for China overall and the Central region, and the spatial error model (SEM) was chosen for the Eastern and Western Regions.

The regression results show that, first, there was absolute  convergence in LRRR in all regions of China. The convergence coefficients for overall China and the Eastern, Central, and Western Regions were all significant at the 1% level, indicating that LRRR converged to respective steady-state levels in the long run when the effects of a range of economic and social factors on LRRR were not considered.

Second, there were differences in the rate of convergence of LRRR across regions. The convergence rates were 2.6%, 1.8%, 1.9%, and 3% for China overall and the Eastern, Central, and Western Regions, respectively (See Table 6.).

Third, China overall and all the regions exhibited different spatial effects. Spatial lags of independent and dependent variables existed in China overall and the Western Region, and both  and  for each model were positively significant at least at the 5% level of significance, indicating that the rates of change in LRRR in the provinces within the regions were simultaneously affected by positive spatial spillovers from the rates of change in other provinces. Spatial error lags existed in the Eastern and Central Regions, and the  in the models for both regions were significant at the 1% level, indicating that the error terms in the spatial regressions of LRRR in one province within the region were spatially correlated and the error terms had spatial effects on other provinces within the region. However, the above analysis on the absolute  convergence of LRRR across the regions was conducted under the assumption that the levels of economic development, industrial structure, urbanization rate, and marketization degree were similar across regions. However, this is not the case in reality, and therefore further conditional  convergence studies are needed. …”

  • The sections are: Introduction, materials and methods, results and discussion, conclusions and policy implications and each section is well covered. I detected no weaknesses, only strengths in the paper.

Response: Thank you very much for your approval of our manuscript.

Reviewer 2 Report

The article entitled “Spatio-temporal variation and decomposition analysis of livelihood resilience of rural residents in China” was reviewed.

The article is well-written, relevant and original. The background is described in detail in the Introduction chapter. The definitions are correct, it was a good idea to give the definitions right at the start.

The paper incorporates a wealth of sources and discusses the literature effectively. There are no major language errors.

The reviewer’s recommendations and comments are below:

The Literature chapter should be separate from the introduction. A short introduction about the relevance of the topic is necessary, and a clearly formulated objective and research question is necessary. The objectives of the study and the research question should be more pronounced, as this would help guide the reader’s frame of mind and would better focus the readers’ attention to the importance of the topic. The reader needs to know why the paper is relevant and what implications can be made. The Literature chapter should be completely separated.

The research design is innovative and appropriate. The method is adequately described, and the results are clearly presented.

It is worth considering how the Material and methods chapter could be made easier to read. This part should be shortened or summarised as the journal is read by Readers (a general audience) who might not be familiar with the used methods.  

It is important to explain in the Conclusion chapter how the results are relevant to the reader, how the method used could be applied in a different setting/different countries. In the Conclusion chapter, the research question(s) should be answered – this is why it must be clearly formulated.

The authors are asked to add source information below the tables and figures (e.g., in Line 404). Source information must be added, it is crucial. The titles of the tables should be formatted according to the Author Guide of the journal. It is important to refer to the tables in the text (e.g., See Table x.).

Please check the Notes of Table 7.  Did you mean 10% significant levels?

Line 692: the policy implications – these could be in a separate chapter.

The reviewer hopes that the comments will contribute to improving the paper.  

Author Response

      Thank you very much for giving us the precious opportunity to revise the manuscript. I had revised the manuscript in light of your comments after a careful discussion with the co-authors. The following are our detailed point-by-point answers to your queries (Some symbols cannot be presented in the web version of the reply, please see the attachment.) We hope the revised manuscript can comply with the high standard of papers accepted for publication.

Response to Reviewer #2

The article entitled “Spatio-temporal variation and decomposition analysis of livelihood resilience of rural residents in China” was reviewed. The article is well-written, relevant and original. The background is described in detail in the Introduction chapter. The definitions are correct, it was a good idea to give the definitions right at the start. The paper incorporates a wealth of sources and discusses the literature effectively. There are no major language errors.

Response: Thank you very much for your approval of our manuscript. The following are our detailed point-by-point answers to your queries.

The reviewer’s recommendations and comments are below:

  • The Literature chapter should be separate from the introduction. A short introduction about the relevance of the topic is necessary, and a clearly formulated objective and research question is necessary. The objectives of the study and the research question should be more pronounced, as this would help guide the reader’s frame of mind and would better focus the readers’ attention to the importance of the topic. The reader needs to know why the paper is relevant and what implications can be made. The Literature chapter should be completely separated.

Response and revision: Thank you very much for your suggestion. We have revised the manuscript accordingly. We have separated the Literature Review chapter from the Introduction chapter, and we have added an explanation of the research purpose, the objectives of the topic, and the importance of the research question in the Introduction chapter. The detailed revision is as follow:

“Livelihoods refer to the means and ways in which people use the resources avail-able around them to maintain and enhance their basic living conditions [1,2]. To a certain extent, livelihood can represent the ability of residents to resist risks and external shocks. This means that setting indicators to quantitatively measure the ability of local residents to adapt and transform in the changing social environment and natural environment has important practical significance for promoting the sustainability of residents' livelihoods. The livelihood resilience can describe the resilience of residents in production and life after external shocks. Therefore, in the case of changes in the social environment and frequent natural disasters, livelihood resilience has gradually become an important reference standard for measuring the quality of life of local residents. At the same time, due to the unbalanced economic development between urban and rural areas in China, there will be obvious regional differences in the livelihood resilience of residents.

As urbanization continues to soar in China, urban and rural areas have different development paths in terms of economic structures and social networks, and therefore, urban and rural residents in China also have varied access to livelihoods [3]. While urban residents mostly rely on a combination of business, social security, and urban public services, which have high stability and sustainability, rural residents tend to sustain their livelihoods by using natural endowments, fixed assets, and social network resources [3,4]. Compared to rural residents, urban residents have higher levels of livelihood and better livelihood security [5]. This is mainly because, firstly, rural residents mainly depend on agricultural income and the agricultural economy is extremely susceptible to environmental changes and human activities [6,7]. In general, rural residents have lower incomes, less social capital, and are less advantaged in the face of abrupt political or economic changes and devastating natural disasters than urban residents, who have more opportunities to education and employment [4,8]. Secondly, economic and social policies are mostly formulated and implemented with a bias towards urban development. These policies mostly favor urban residents rather than rural residents, resulting in a huge gap between urban and rural residents in terms of income and social security [9].

Meanwhile, most local governments' fiscal expenditures tend to be constructive expenditures that promote economic growth in the short term, while livelihood expenditures such as education and health care, which have a longer investment return cycle, are downplayed [10]. Biased economic and social policies and the fiscal expenditure structure of local governments have resulted in a wide gap between urban and rural basic public services, mainly in the form of poorer infrastructure and inadequate public service provision to protect people's livelihoods in rural areas [5,11]. In addition, due to limited access to public resources, rural residents recover more slowly after natural disasters [7]. Thus, compared to urban residents, rural residents are disadvantaged in terms of livelihood capacity. However, although livelihood capacity can represent to some extent the ability of rural residents to withstand risks and external shocks, it cannot fully portray the ability of rural residents to recover in production and life after being exposed to external shocks. Since the concept of livelihood resilience represents the ability to adapt and change in the changing society and natural environment, the livelihood resilience of rural group is more worthy of attention.

One of China's main challenges in implementing the "Rural Revitalization" strategy is how to construct better and more sustainable villages in the midst of urbanization [12]. Rural dwellers have external effects that go beyond only the environmental change, these impacts also include social and economic developments. For sustainable rural development, it is crucial to quantify how well rural population can adapt to changes in social and natural environment. On this basis, this paper will offer policy suggestions for promoting the sustainability of rural residents' livelihoods and reducing regional differences by optimizing the index system and clarifying the dynamic changes and regional differences of livelihood resilience of rural population.”

Moreover, we also discussed more findings of previous studies in the Literature Review chapter, as follows:

“The literature shows that the dynamics of the livelihood resilience of rural residents (LRRR) are highly correlated with external pressures on rural residents, and related policies. First, changes in the natural environment significantly affect LRRR. Some studies have found that frequent natural disasters such as earthquakes, floods, and droughts can reduce LRRR and widen the gap in livelihood resilience among rural residents [25,26]. However, by lowering agricultural productivity and eroding resistance, severe climate change can also limit LRRR [27,28]. Second, LRRR is highly correlated with regional macro policies. To address the multiple dilemmas of post-disaster reconstruction, environmental protection, improvement of human well-being, and alleviation of poverty, governments have targeted policies such as environmental migration and poverty alleviation relocation [2,29]. Among them, the environmental migration policies can improve the natural capital of rural residents, and the poverty alleviation relocation policies can improve the physical capital and human capital. Therefore, most of the policies can improve LRRR to varying degrees [30-32]. From these studies, it is easy to find that properly defining the meaning of LRRR, comprehensively measuring its level, and portraying its evolution pattern are not only key prerequisites for promoting the development of rural residents' livelihood, but also important and realistic references for assessing the effects of implementing existing relevant policies.

Numerous studies have analyzed the measurement of LRRR, but there is still no agreement over the optimal composition of its indicators. Most of the literature has characterized the idea of livelihood resilience in terms of two key issues: capacity and capital. Focusing on the capability perspective, Speranza et al. [22] developed a three-dimensional framework for analyzing livelihood resilience. The framework includes three types of indicators: buffer capacity, self-organization capacity, and learning capacity. Several studies have applied this analytical framework to examine changes in LRRR and concluded that buffer capacity is a key factor in maintaining the resilience of rural residents' livelihoods [33,34]. Based on this, Quandt [21] drew on the sustainable livelihoods approach to further subdivide buffer capacity from a capital perspective. Capital was decomposed into financial capital, human capital, social capital, physical capital, and natural capital. This type of livelihood construction focuses more on the material value of capital [3,35], arguing that livelihood strategies and levels of well-being are heavily influenced by various levels of and access to livelihood capital [1,36]. Among them, for rural residents, social capital is more effective in coping with external changes in rural network relations [37]. Other literatures have studied the relationship between livelihood and risk based on this framework, and found that both environmental risk and disease risk have a significant negative impact on achieving sustainable livelihoods for rural residents [38]. The above studies have mainly focused on the livelihood resources available to rural residents, but less attention has been paid to the impact of rural residents' interactions with the external environment on livelihood resilience. However, some scholars have constructed a new re-search framework that incorporates the adaptive capacity [23,39] or disaster resilience [7,40] of rural residents, arguing that the ability to choose livelihood strategies in response to changing external conditions is equally critical. On this basis, LRRR could be measured through disturbance, sensitivity, and adaptability at farm level [41].”

  • The research design is innovative and appropriate. The method is adequately described, and the results are clearly presented.

Response: Thank you very much for your approval of our manuscript.

  • It is worth considering how the Material and methods chapter could be made easier to read. This part should be shortened or summarised as the journal is read by Readers (a general audience) who might not be familiar with the used methods.

Response and revision: Thank you very much for your comment. We have shortened some of the content of the Material and methods chapter and the revision is as follows:

“In equation (9),  and . In equations (11) and (12),  and .  is the relative impact of the LRRR between the regions  and , and is calculated as shown in the following equations: …”

“…Based on the results of kernel density estimation, the characteristics of changes in LRRR can be observed in the curve images. The horizontal position of the kernel density curve of a single period can represent high or low LRRR. The height and width of the curve peaks can reflect the degree of aggregation of LRRR within the region. The number of peaks can portray the degree of polarization of the LRRR, and the distribution extension, i.e., the degree of trailing, can describe the distance between the provinces with the highest or lowest LRRR and other provinces, where more severe trailing represents higher intra-regional variation.”

“…This study considered the possible spatial correlations in LRRR arising from the acceleration of resource flows and the enhancement of inter-regional interaction effects [63]. Based on this, a neighbouring spatial weight matrix was constructed and a dynamic spatial panel model was applied to identify the  convergence characteristics of LRRR in three major regions of China. Common spatial econometric models include spatial autoregressive model (SAR), spatial error model (SEM), spatial Durbin model (SDM), etc. [64]. However, since the spatial correlation in LRRR in China is not known, this study fitted each of the above three models to select a suitable model for convergence analysis. …”

  • It is important to explain in the Conclusion chapter how the results are relevant to the reader, how the method used could be applied in a different setting/different countries. In the Conclusion chapter, the research question(s) should be answered – this is why it must be clearly formulated.

Response and revision: Thank you very much for your kindly and useful comment. We have added some explanations accordingly in Conclusion chapter as follows:

“Academic research has recently concentrated on ways to make LRRR sustainable. Improving LRRR is not only a key issue in China's current rural construction and governance and the realization of the "Rural Revitalization" strategy. For many developing countries, improving LRRR is also of great value in advancing modernization. The research model developed in this work advances the dynamic comprehension of LRRR and offers suggestions for the optimization and calculation of the index system measuring the LRRR in developing nations. The research findings also offer developing nations policy guidance for achieving the sustainable livelihood resilience.”

  • The authors are asked to add source information below the tables and figures (e.g., in Line 404). Source information must be added, it is crucial. The titles of the tables should be formatted according to the Author Guide of the journal. It is important to refer to the tables in the text (e.g., See Table x.).

Response and revision: Thank you very much for your helpful suggestion. First, we have added source information below the tables and figures. Second, the titles of the tables have been formatted according to the Author Guide of the journal as follows:

“Table 1. Definitions and units of measurement of the indicators of the livelihood resilience of rural residents.

Table 2. Descriptive statistics of variables.

Table 3. Livelihood resilience of rural residents in China by province.

Table 4. Characteristics of the dynamic distribution of the livelihood resilience of rural residents in China.

Table 5  convergence of the livelihood resilience of rural residents in China by region.

Table 6. Absolute  convergence of the livelihood resilience of rural residents in China by region.

Table 7. Conditional  convergence of the livelihood resilience of rural residents in China by region.”

Moreover, in the revised manuscript, we have referred the tables in the text as follows:

“…Second, there were differences in the rate of convergence of LRRR across regions. The convergence rates were 2.6%, 1.8%, 1.9%, and 3% for China overall and the East-ern, Central, and Western Regions, respectively (See Table 6.) …”

“…The convergence rates were 3.3%, 2.3%, 1.9%, and 5.1% for China overall and the Eastern, Central, and Western Regions, respectively (See Table 7.) …”

  • Please check the Notes of Table 7. Did you mean 10% significant levels?

Response and revision: Thank you very much for your helpful suggestion. In tables 6 and 7, the regression results are at least significant at the 5% level, so we have deleted “10% significant levels” in the notes. The revised notes of tables 6 and 7 as follows:

“Notes: Standard errors are in parentheses. ***, ** denote significance at the 1% and 5% levels, respectively. Source: authors’ own calculation.”

  • Line 692: the policy implications – these could be in a separate chapter.

Response and revision: Thank you very much for your helpful comment. We have made the Policy Implications a separate chapter, and the revision as follows:

“Based on the above findings, we draw the following policy implications: First, it is important to fully understand the reality that the LRRR in China is on the decline. Although the LRRR in the Western Region had increased, the overall LRRR in China was still declining, and there is much room for improvement. We should fully under-stand the important role of livelihood resilience in sustainable rural development, improve rural public service infrastructure, reduce the vulnerability of rural residents to external shocks, and enhance their adaptability. Second, the spatially uneven distribution of LRRR in China should be fully recognized. The difference in LRRR between provinces in China overall and the Central Region has narrowed, while the difference in LRRR between provinces in the Eastern and Western Regions has increased. While promoting the steady improvement of rural residents' livelihood resilience, we should continue to strengthen the awareness of regional spatial synergy and explore the rea-sons for the existence and expansion of regional differences. At the same time, with the goal of balanced development, equal development opportunities for rural residents in all regions, especially disadvantaged regions, should be guaranteed. The differences in livelihood resilience among rural residents in the Eastern, Central, and Western Regions was the main source of the differences in the LRRR in China as a whole. There-fore, when formulating relevant policies, policy preferences can be given to regions with lower livelihood resilience to reduce regional differences. Third, the convergence characteristics of rural residents' livelihood resilience suggest that, while reducing the inter-regional differences among the Eastern, Central, and Western Regions, it is important to pay attention to the coordination of the convergence rate of rural residents' livelihood resilience among regions. Meanwhile, provinces with higher LRRR should make use of the spatial spillover effect, amplify the radiation effect to neighboring areas, and simultaneously raise the livelihood resilience in neighboring provinces.”

  • The reviewer hopes that the comments will contribute to improving the paper.

Response: Thank you very much for your helpful and constructive suggestions.
